# Hierarchical Classification by Training to Diffuse on the Manifold

## Abstract

Hierarchical classification, the problem of classifying images according to a hierarchical taxonomy, has practical significance owing to the principle of "making better mistakes", i.e., better to predict correct coarse labels than incorrect fine labels. Nevertheless, the literature does not sufficiently study this problem, presumably because using top-1 accuracy to benchmark methods tends to yield a ranking order consistent with those using hierarchical metrics. On the other hand, for a downstream task of classification, today's *de facto* practice is to *finetune* a pretrained deep neural network using the cross-entropy loss on leaf classes, resulting in a leaf-class softmax classifier which even rivals sophisticated hierarchical classifiers atop deep nets. We argue that hierarchical classification should be better addressed by regularizing finetuning with explicit consideration of the given hierarchical taxonomy, because data intuitively lies in hierarchical manifolds in the raw feature space defined by the pre-trained model. To this end, we propose a hierarchical cross-modal contrastive loss that computes contrative losses w.r.t labels at hierarchical levels in the taxonomy (including both hierarchy and text concepts). This results into features that can better serve hierarchical classification. Moreover, for inference, we re-conceptualize hierarchical classification by treating the taxonomy as a graph, presenting a diffusion-based methodology that adjusts posteriors at multiple hierarchical levels altogether. This distinguishes our method from the existing ones, which are either top-down (using coarse-class predictions to adjust fine-class predictions) or bottom-up (processing fine-class predictions towards coarse-label predictions). We evaluate our method by comparing them against existing ones on two large-scale datasets, iNat18 and iNat21. Extensive experiments demonstrate that our method resoundingly outperforms prior arts w.r.t both top-1 accuracy and hierarchical metrics.

## 1 Introduction

Hierarchical classification (Naumoff, 2011; Deng et al., 2012; Zhu & Bain, 2017; Bertinetto et al., 2020) has long been a pivotal and challenging problem in machine learning. It aims to categorize images w.r.t a given hierarchical taxonomy, adhering to the principle of "making better mistakes" — essentially, favouring correct coarse-class predictions over inaccurate fine-class predictions (Deng et al., 2012; Wu et al., 2020).

Methods of hierarchical classification improve either training or inference. Existing inference methods can be divided into two types: top-down (Redmon & Farhadi, 2017), and bottom-up (Valmadre, 2022). Top-down methods adjust the posterior for predicting a specific class by using its parent/ancester posterior probabilities. They often underperform bottom-up methods Redmon & Farhadi (2017); Bertinetto et al. (2020), which prioritise predicting the leaf-classes and subsequently calculate posteriors for the parent/ancestor classes. Valmadre (2022) attributes the underperformance of top-down methods to the high diversity within coarse-level categories, soliciting effective training methods. Perhaps surprisingly, although these sophisticated hierarchical classification methods show promising results in certain metrics, they do not consistently rival the simplistic flat-softmax baseline, which learns a softmax classifier on the leaf classes only. The status quo leads to a natural question: *Is it still helpful to make predictions for hierarchical classes other than the leaf classes for better hierarchical classification?* That said, it is still an open question how to effectively exploit hierarchical taxonomy to improve training and inference for hierarchical classification.

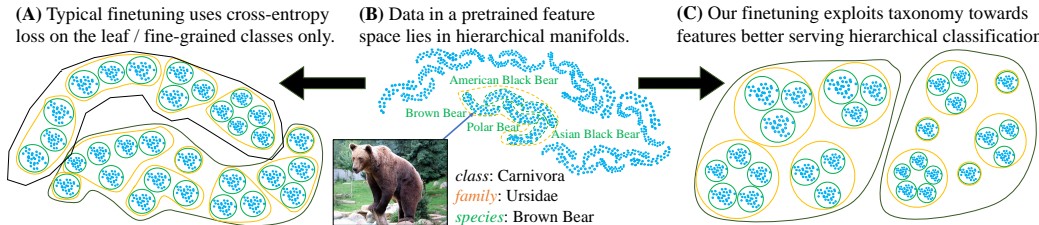

**(A)** Typical finetuning uses cross-entropy loss on the leaf / fine-grained classes only.

**(B)** Data in a pretrained feature space lies in hierarchical manifolds.

**(C)** Our finetuning exploits taxonomy towards features better serving hierarchical classification.

Figure 1: To solve a downstream task of classification, a *de facto* practice is to fine-tune a pretrained model using the cross-entropy loss on leaf classes (e.g., Brown Bear at the species level). **(A)**: This yields features that help leaf-class classification but fail to model their hierarchical relationships w.r.t a taxonomy (e.g., Ursidae at the family level). Hence, it does not necessarily help hierarchical classification. Nevertheless, such features are better than the "raw features" of the pretrained model, which provides a feature space **(B)** where data hypothetically lie in hierarchical manifolds w.r.t the taxonomy. **(C)**: Differently, we propose to fine-tune the pretrained model by *explicitly* exploiting the hierarchical taxonomy towards features that can better serve the task of hierarchical classification (Fig. 2).

We argue that, to better approach hierarchical classification for a downstream task that defines a hierarchical taxonomy, one should first explicitly exploit this taxonomy to learn features (Fig. 1), not just finetuning a pretrained model using the cross-entropy loss on leaf classes only (Bertinetto et al., 2020). Note that a taxonomy contains not only a hierarchy of concepts (e.g., species, genus, order, family, etc.) but also describable texts or names for the concepts. This motivates us to finetune a pretrained vision-language model (VLM) (Radford et al., 2021; Wang et al., 2023; Goyal et al., 2023). For better finetuning, we introduce a hierarchical cross-modal contrastive fine-tuning strategy (HCCF) (Sec. 3.2). HCCF explicitly exploits hierarchical taxonomy towards learning better features, which directly mirror the given taxonomy and hence better serve hierarchical classification.

Moreover, we argue that one should also collectively adjust posteriors at multiple hierarchical levels towards the final results of hierarchical classification. To this end, we present a set of diffusion-based methods for inference (Sec. 3.3), inspired by the literature of information retrieval Page et al. (1998); Iscen et al. (2017); An et al. (2021) which shows that diffusion is adept at mapping manifolds. This distinguishes our methods from existing top-down and bottom-up inference approaches that linearly interpret hierarchical classification. Our methods treat the hierarchical taxonomy as a graph, enabling probability distribution in the taxonomy. To the best of our knowledge, our work makes the first attempt to apply diffusion to hierarchical classification. Extensive experiments demonstrate that our diffusion-based inference methods, along with HCCF, achieve state-of-the-art performance and resoundingly outperform prior arts (Sec. 4.2).

To summarize, we make three major contributions.

1. We revisit the problem of hierarchical classification from the perspective of manifold learning, offering new insights in the contemporary deep learning land.

2. We present the hierarchical cross-modal contrastive finetuning strategy for finetuning a model to better solve the problem of hierarchical classification.

3. We introduce a novel diffusion-based inference methodology to exploit posteriors at multiple levels towards the final prediction.

## 2 RELATED WORKS

**Hierarchical classification**. Hierarchical classification holds significance, ensuring broader-level results even when detailed predictions are elusive. Datasets like ImageNet (Russakovsky et al., 2015) and WordNet (Miller, 1995) have long emphasized taxonomy, while newer ones like iNat18 (Van Horn et al., 2018) and iNat21 offer finer-grained labels. Research in this domain is robust, with seminal works like "Hedging Your Bet" (Deng et al., 2012) and contemporary deep learning approaches employing flat softmax, oftmargin, and descendant softmax training losses (Valmadre, 2022), along with bottom-up (Valmadre, 2022) and top-down (Redmon & Farhadi, 2017) inferences. Its practical applications are evident in areas like long-tailed 3D detection for autonomous driving (Peri et al., 2023), emphasizing specific metrics, methods, and joint training. Despite ex-

tensive research, recent findings suggest that advanced training and inference methods don't always surpass the flat softmax baseline (Valmadre, 2022). This paper presents innovative techniques that harness hierarchical data more efficiently during both the training and inference stages.

**Long-tailed recognition** (LTR). Long-tail categorization is an active research topic, as the long-tail feature is prevalent across coarse-level, fine-grained, and instance-level categorizations. Current strategies often employ data rebalancing (Mahajan et al., 2018; Chawla et al., 2002) or class-balanced loss functions (Cao et al., 2019) to improve the classification accuracy of infrequent classes. Despite these advancements, the exploration of the long-tail attribute within hierarchical categorization remains less investigated, indicating a need for further research in this area.

**Fine-grained visual categorization** (FGVC). Fine-grained categorization, a task bridging coarse-level classification and instance-level classification, presents both significant value and substantial challenges (Akata et al., 2015; Yang et al., 2018). In cases where predicting the fine-grained level tag proves difficult, users often still prefer an accurate coarse-level result, highlighting the importance of hierarchical research within the fine-grained classification (Deng et al., 2012). This paper contributes to this aspect, pushing forward the understanding and application of hierarchical fine-grained categorization in the context of long-tail distributions.

**Diffusion**. Diffusion is an advanced methodology adept at faithfully delineating the manifold within a data distribution by leveraging the interconnectedness inherent in a Markov chain (Zhou et al., 2003a;b). A renowned variation of this method, PageRank (Page et al., 1998), has achieved considerable success in various business endeavors. Moreover, it has been extensively employed in the realm of image retrieval (Iscen et al., 2017; An et al., 2021), an application of instance-level classification. However, its potential in broader classifications, such as fine-grained and hierarchical categorizations, has not been extensively researched. In this paper, we pioneer the exploration of its utility in understanding and utilizing the relationships within these broader, fine-grained, and hierarchical classifications.

# 3 METHODS

**Hierarchical classification and notations**. This paper delves into the intricacies of Single-Path Labels (SPL) and Non-Mandatory Leaf-Node Prediction (NMLNP) in hierarchical classification. In SPL, a sample is restricted from belonging to multiple distinct classes unless there exists a superclass-subclass relationship. On the other hand, NMLNP allows the classifier to predict any class within the hierarchy, not being confined to just the leaf nodes. In this study, we let $Y$ denote the entirety of categories within the taxonomy tree. For a given node $y \in Y$, $C(y)$ signifies its child nodes, while $A(y)$ stands for its ancestor nodes. The set of leaf nodes is represented by $L$.

## 3.1 HIERARCHICAL MANIFOLD

We introduce a hierarchical manifold model in the embedding space to elucidate the intricacies of hierarchical classification. Although data manifolds are prevalent in high-dimensional spaces, what sets hierarchical classification apart is its distinct manifold structure. As depicted in Fig 1, before optimization, each category in the embedding space can be visualized as a separate manifold. Drawing an analogy to the parent-child node relationship, **parent manifolds envelop child manifolds**. An optimally refined embedding space should discern manifolds across all hierarchical levels.

The hierarchical manifold assumption holds merit. Given that manifolds are frequently observed in diverse real-world datasets, it's plausible that the embedding space houses these hierarchical manifolds prior to achieving an optimal training solution. This sheds light on the limitations of current techniques in addressing the hierarchical classification challenge. As illustrated in Fig. 1, existing methods, **failing to grasp the nuances of higher-level manifolds**, might misclassify an image under the family level, even if they correctly identify it at the species level.

While there are extant hierarchical loss functions aimed at this problem, they predominantly predict only the leaf node categories. Consequently, the hierarchical loss equation ultimately converges to supervision solely at the leaf level. For instance, when employing bottom-up inference for interior

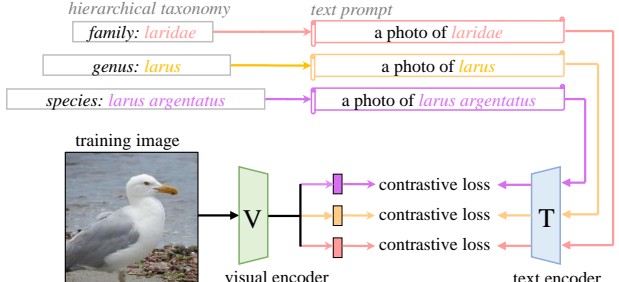

Figure 2: The proposed Hierarchical Cross-modal Contrastive Finetuning (HCCF) exploits hierarchical taxonomy to adapt a pretrained visual encoder to the downstream task of hierarchical classification. It sums contrastive losses between a training image and its taxonomic names at multiple levels. To the best of our knowledge, we make the first attempt to fine-tune a vision-language model using a predefined taxonomy for hierarchical classification.

node prediction results as:

$$
q_y(\theta) = \begin{cases} [\text{softmax}_L(\theta)]_y & \text{if } y \in L \\ \sum_{v \in C(y)} q_v(\theta) & \text{if } y \notin L \end{cases}
\tag{1}
$$

The negative log-likelihood concerning the interior nodes **is reduced to the leaf nodes** as $\ell(y, \theta) = -\log q_y(\theta) = -\log\left(\sum_{u \in L(y)} \exp \theta_u\right) + \log\left(\sum_{u \in L(y)} \exp \theta_u\right)$. Advanced losses, such as soft-margin and descendant softmax (Valmadre, 2022), also focus on the leaf level, neglecting the separation of higher-level manifolds. This results in suboptimal outcomes for hierarchical classification.

The hierarchical manifold model inspires novel strategies for both training and inference. For the training phase, the model suggests that we should: 1) **Effectively leverage the multiple labels** associated with each training image, and 2) **Enhance the separation** between sample distributions from different categories across various levels in the embedding space, thereby reducing misclassification risks. During inference, the model motivates us to use **diffusion—a technique renowned for its efficacy with manifolds**—to refine the scores predicted by the neural network.

### 3.2 HIERARCHICAL CROSS-MODAL CONTRASTIVE FINE-TUNING

To more **effectively map the taxonomy relations in the embedding space**, we initially employ the Vision-language pretrained model, CLIP (Radford et al., 2021), as our primary visual encoder. Using textual descriptions for each image provides a more comprehensive supervisory signal, capturing both leaf and interior node relationships in the taxonomy tree. While CLIP's superiority over ImageNet as a pretrained model is somewhat recognized, its efficacy in hierarchical classification remains untested. Our experiments on the renowned iNat18 dataset (Van Horn et al., 2018) indicate significant improvements (Table 1).

Our advancements extend beyond the utilization of the CLIP pre-trained model. We propose a hierarchical cross-modal contrastive loss, aiming to **extend the distance** between sample distributions across varied categories and levels (shown in Fig. 2). This strategy is anchored in two core tenets of our hierarchical manifold model. Firstly, we harness the full potential of textual descriptions for each training image. By employing the CLIP text encoder, we encode the hierarchical labels of these images. Distinct from prevailing hierarchical losses, our interior node prediction isn't merely inferred from leaf nodes. Instead, it's directly guided by the embedding vectors of text labels across different levels, enabling a more nuanced understanding of category relationships and better capturing of higher-level manifolds. Secondly, our methodology employs contrastive loss, ensuring maximal separation between samples from diverse categories, thereby mitigating the complexities introduced by hierarchical manifolds. Our hierarchical cross-modal contrastive fine-tuning loss is defined as:

$$
L(f, g) := \sum_{l=1}^{L} \left( \sum_{i=1}^{N} -\log \frac{\exp(\bar{f}^l(I_i) \cdot \bar{g}(T_i^l))}{\sum_{j=1}^{N} \exp(\bar{f}^l(I_i) \cdot \bar{g}(T_j^l))} + \sum_{i=1}^{N} -\log \frac{\exp(\bar{f}^l(I_i) \cdot \bar{g}(T_i^l))}{\sum_{j=1}^{N} \exp(\bar{f}^l(I_j) \cdot \bar{g}(T_i^l))} \right),
\tag{2}
$$

where $\bar{f}^l(I_i)$ is normalized embedding of the $i$-th image $I_i$ from the visual encoder $f^l$, which consists of visual backbone and level-specific head. $\bar{g}(T_j^l)$ is the normalized text embedding of the text $T_j^l$, that is the $j$-th sample of level $l$ extracted from text encoder $g$. Assuming there are $N$ image-texts pairs in one batch, $I_i$ is input image and $T_i^l$ denotes the ground truth label at level $l$. All

taxonomy tree text prompts utilize a shared text encoder, mitigating overfitting risks and conserving training and inference resources. The visual encoder comprises a shallow feature extractor and a level-specific extractor head for every level, ensuring encoding aligns with the hierarchical taxonomy level. Both visual and text encoders are updated during training, and text encoding of every taxonomy level serve as linear classifier weights during inference.

## 3.3 DIFFUSION-BASED INFERENCE

Through our new training strategy, we generate prediction scores for all taxonomy categories. The ensuing challenge is to utilize these scores for inference and robust decision-making effectively.

**Existing inference techniques**, namely the top-down (Redmon & Farhadi, 2017) and down-top (Valmadre, 2022) approaches, can be further improved. The top-down method computes the conditional likelihood of each child node based on its parent nodes. While theoretically appealing, it is empirically outperformed by the down-top approach (Redmon & Farhadi, 2017; Bertinetto et al., 2020; Valmadre, 2022). Valmadre (Valmadre, 2022) attributes this underperformance to the high diversity within coarse-level categories and advocates using fine-grained scores to infer hierarchical outcomes. We align with Valmadre's observations, yet we assert that predictions for mid-level categories have inherent value when utilizing our innovative diffusion-based inference.

**Motivation**. When a category receives an anomalously high or low score from the neural network, we can recalibrate this score based on the scores of its neighboring categories within the taxonomy tree. Essentially, sub-categories under the same parent category should exhibit consistent scoring patterns, either high or low. By diffusing the scores across the taxonomy's structural connections to achieve equilibrium, we can enhance the initial predictions made by the neural network Remarkably, experiment results show that our method enhances both the leaf-level top-1 accuracy and the overall hierarchical performance, outperforming existing techniques (Sec. 4.4).

**Notation**. Given a total of $n$ categories (including intermediate categories) in the taxonomy graph, we define a connection matrix $W \in R^{n \times n}$ to describe the interrelationships among categories within the graph. Let $f^0 \in R^n$ be the prediction output of the neural network. Our target is to refine $f^0$ based on $W$ to get the final $f^\star$, which gives both better leaf-level and hierarchical performance.

**Connection matrix**. We first use the expert-designed taxonomy given by each dataset to define the connection matrix $W$. That is, $w_{ij} = 1$ if category $i$ and $j$ have the parent-children relation in the taxonomy tree. Otherwise $w_{ij} = 0$. Here, we assume the graph is undirected, and the connection matrix is symmetric ($W = W^T$). The self-similarity is set as zero ($\mathrm{diag}(W) = \mathbf{0}$). We will explore more weight options within this matrix in subsequent sections.

Normalization for the connection matrix is an essential step for diffusion in information retrieval. We find it is also necessary in the hierarchical classification. In this paper, we use the symmetrically normalization as follows:

$$W_n = D^{-1/2} W D^{-1/2}, \quad D = \mathrm{diag}(W \mathbf{1}_n). \tag{3}$$

**Iteration**. Our diffusion mechanism iteratively updates the category scores according to the following:

$$f^{t+1} = \alpha W_n f^t + (1 - \alpha) f^0, \tag{4}$$

where $\alpha$ is set among $(0, 1)$. This is a "random work" algorithm in the taxonomy graph. Intuitively, for each iteration, each category spreads its prediction score to its neighbor categories with probability $\alpha$, and follows the initial neural network prediction with probability $1 - \alpha$.

**Convergence**: The iterative process is assured to converge towards a stationary distribution (Zhou et al., 2003b). We provide a straightforward proof here. By recursively integrating $f^1 = \alpha W_n f^0 + (1 - \alpha) f^0$ into subsequent iterations $f^2$, $f^3$, and so on, we derive:

$$f^t = (\alpha W_n)^t f^0 + (1 - \alpha) \sum_{i=0}^{t} (\alpha W_n)^i f^0. \tag{5}$$

As $t$ approaches infinity, the term $(\alpha W_n)^t$ approaches zero, and the summation term converges to $(I - \alpha W_n)^{-1}$, where $I$ denotes the identity matrix of size $n$. Thus, the eventual stationary distribution is expressed as:

$$f^* = (1 - \alpha)(I - \alpha W_n)^{-1} f^0. \tag{6}$$

**Relation to Spectral Clustering**: It's pertinent to elucidate the connection between our hierarchical classification diffusion and spectral clustering, given that both methodologies emphasize node grouping within a graph. Notably, the term $(I - \alpha W_n)$ in Equation 6 can be interpreted as a variant of the symmetrically normalized Laplacian $(I - W_n)$ for the taxonomy graph. This Laplacian is instrumental in spectral clustering, enabling the capture of the data's intrinsic topological characteristics. In the spectral clustering paradigm, each node is characterized by a k-dimensional spectral space vector, derived from the k eigenvectors satisfying $v = (I - W_n)^{-1} \lambda v$. Conversely, our diffusion process assigns each node a singular scalar score, as dictated by Equation 6. Conceptually, our diffusion approach can be perceived as a tailored spectral clustering for the neural network's predicted vector $f^0$, pinpointing a category subset with peak scores in the spectral domain.

**Differentiable diffusion**: As demonstrated in Eq. 6, the diffusion process converges to a closed form. Intriguingly, this represents a linear transformation from the initial scores $f^0$ to the final state $f^\star$. Currently, the connection matrix $W$ is constructed based on the provided taxonomy tree structure, comprising binary values that might not accurately capture the genuine relationships between category pairs. Given a substantial sample size from the training set, we investigate the potential of training a linear mapping directly to supplant the closed form. This differentiable method could offer a more nuanced understanding of the relationships between categories. We call this new approach differentiable diffusion.

Our main contribution lies in introducing **an advanced diffusion method**, specially designed for using the taxonomy graph's structure. To the best of our knowledge, this is the first work to apply diffusion techniques to hierarchical classification problems. While existing literature has extensively explored the diffusion of instance space (like web and image) with considerable success (Page et al., 1998; An et al., 2021), the impact of diffusion on the category space (how to group the instances) remains largely uncharted territory. This diffusion approach offers several distinct advantages over existing top-down and down-top inference:

1. **Comprehensive graph utilization:** Unlike traditional methods that focus solely on direct parent-child relationships, our diffusion technique leverages the entire graph structure, including sibling relationships.

2. **Iterative information blending:** While existing methods transfer information once through the graph edge, our diffusion process iteratively blends information at each node until a stable state is achieved, thereby maximizing the utility of all predicted category nodes.

3. **Manifold problem resolution:** Our method addresses the manifold problem by utilizing inter-category relationships, on which we elaborate subsequently.

## 4 EXPERIMENTS

### 4.1 IMPLEMENTATIONS

To assess the efficacy of our novel training and inference approach for hierarchical classification, we employ the metrics and dataset from the recent study by Valmadre (Valmadre, 2022). This study presents state-of-the-art (SOTA) methods, comprehensive experiments on existing techniques, and a suite of robust metrics tailored for hierarchical classification. Similar to Valmadre (2022), all the experiments use ResNet 50 (He et al., 2016) as the backbone. Valmadre's benchmark dataset is the balanced iNaturalist 21-mini (iNat21). In our evaluation, we extend the datasets to include iNaturalist 18 (iNat18), showcasing the versatility of our method and its performance under long-tailed distributions. In line with Valmadre's approach (Valmadre, 2022), our metrics are derived from operating curves, encompassing Average Precision (AP), Average Correct (AC), Recall at X% Correct (R@XC), and a specificity measure. We also incorporate single prediction metrics such as Majority F1, Leaf F1, and Leaf Top1 Accuracy. Notably, while Leaf Top1 Accuracy gauges leaf-level accuracy, the other metrics focus on hierarchical classification performance. Our methods are benchmarked against various SOTA hierarchical classification techniques, including flat softmax (Bertinetto et al., 2020), Multilabel focal (Lin et al., 2017), Cond softmax (Redmon & Farhadi, 2017), Cond sigmoid (Brust & Denzler, 2019), DeepRTC (Wu et al., 2020), PS softmax (Wu et al., 2020), Softmargin and (Valmadre, 2022) descendent softmax (Valmadre, 2022).

Table 1: Benchmarking results on the iNat18 dataset. We report numbers w.r.t both hierarchical metrics (Valmadre, 2022) and the standard top-1 accuracy on leaf classes (dubbed Leaf Top1 in the last column). Our HCCF, which contrastively finetunes a pretrained model using all the taxonomic levels, significantly outperforms prior arts. Additionally applying diffusion improves performance notably further.

| Model | AP | AC | R@90C | R@95C | Majority F1 | Leaf F1 | Leaf Top1 |
|---|---|---|---|---|---|---|---|
| Flat softmax (Bertinetto et al., 2020) | 61.18 | 58.94 | 45.44 | 37.58 | 64.27 | 64.57 | 47.33 |
| Multilabel focal (Lin et al., 2017) | 46.70 | 43.97 | 34.05 | 28.05 | 50.69 | 49.91 | 14.85 |
| Cond softmax (Redmon & Farhadi, 2017) | 54.13 | 51.12 | 36.68 | 30.07 | 58.74 | 58.60 | 36.94 |
| Cond sigmoid (Brust & Denzler, 2019) | 52.04 | 49.29 | 35.23 | 29.31 | 55.46 | 58.29 | 36.36 |
| Deep RTC (Wu et al., 2020) | 60.07 | 54.25 | 23.69 | 14.33 | 66.72 | 66.72 | 47.13 |
| PS softmax (Wu et al., 2020) | 64.15 | 62.02 | 49.54 | 42.02 | 67.50 | 67.44 | 49.21 |
| Softmargin (Valmadre, 2022) | 58.53 | 55.86 | 40.28 | 33.71 | 58.73 | 63.70 | 45.10 |
| Descendant softmax (Valmadre, 2022) | 61.88 | 59.65 | 46.79 | 38.49 | 65.48 | 65.32 | 48.71 |
| HCCF | _72.75_ | _70.60_ | _59.56_ | _52.60_ | _72.73_ | _75.16_ | _55.78_ |
| HCCF + diffusion | **73.48** | **71.88** | **62.48** | **55.53** | **75.94** | **75.71** | **56.33** |

Table 2: Benchmarking results on the iNat21 dataset. We report numbers w.r.t both hierarchical metrics (Valmadre, 2022) and the standard top-1 accuracy on leaf classes (dubbed Leaf Top1). Our HCCF finetuning, which contrastively finetunes a pretrained model using all the taxonomic levels, significantly outperforms prior arts. Additionally applying diffusion to inference improves performance notably further.

| Model | AP | AC | R@90C | R@95C | Majority F1 | Leaf F1 | Leaf Top1 |
|---|---|---|---|---|---|---|---|
| Flat softmax (Bertinetto et al., 2020) | 66.17 | 64.32 | 53.85 | 47.02 | 68.87 | 68.69 | 50.89 |
| Multilabel focal (Lin et al., 2017) | 54.58 | 50.35 | 36.16 | 30.45 | 50.62 | 60.27 | 31.05 |
| Cond softmax (Redmon & Farhadi, 2017) | 58.88 | 56.26 | 42.95 | 36.23 | 62.85 | 62.80 | 41.64 |
| Cond sigmoid (Brust & Denzler, 2019) | 59.24 | 56.74 | 42.84 | 35.61 | 61.41 | 65.11 | 44.64 |
| Deep RTC (Wu et al., 2020) | 63.92 | 58.07 | 25.36 | 14.10 | 70.17 | 70.22 | 51.43 |
| PS softmax (Wu et al., 2020) | 68.22 | 66.49 | 56.20 | 49.85 | 71.07 | 70.80 | 52.76 |
| descendant softmax (Valmadre, 2022) | 64.95 | 62.71 | 48.84 | 42.59 | 64.64 | 69.03 | 50.55 |
| softmargin (Valmadre, 2022) | 66.53 | 64.72 | 54.41 | 47.91 | 69.39 | 69.09 | 52.22 |
| HCCF (Ours) | _72.46_ | _70.52_ | _60.49_ | _53.66_ | _73.35_ | _74.72_ | _55.11_ |
| HCCF + diffusion (Ours) | **73.16** | **71.62** | **62.81** | **55.97** | **75.31** | **75.32** | **55.86** |

## 4.2 COMPARE WITH STATE-OF-THE-ART METHODS

We performed a comparative analysis of our innovative training and inference methods against established state-of-the-art (SOTA) hierarchical techniques using the iNat18 and iNat21 datasets. Table 1 and Table 2 demonstrate the enhanced performance of our approach over existing SOTA methods across both datasets. Unless otherwise indicated, all methods utilized the same CLIP pretrained model.

In our implementation of the SOTA methods, we strictly adhered to the code provided by Valmadre (Valmadre, 2022). Our results for the iNat21 are consistent with those presented by Valmadre. Although (Valmadre, 2022) did not provide outcomes for iNat18, we included results for this dataset to illustrate our model's capability in handling long-tailed distributions, noting that iNat18 is long-tailed while iNat21 is balanced.

While a direct comparison with (Valmadre, 2022) for iNat18 is not available, we ensured the reliability of our results by using the reproduction code and settings from Valmadre's open-source resources. These results emphasize the advantages of our method over existing SOTA methodologies, proving effective for both balanced and long-tailed datasets. Our hierarchical contrastive training approach sets new standards in the field, outperforming existing SOTA methods for both the iNat18 and iNat21 datasets.

## 4.3 ABLATION STUDY ABOUT HIERARCHICAL CROSS-MODAL CONTRASTIVE FINE-TUNING

We conducted an ablation study to assess the impact of each component in hierarchical cross-modal contrastive fine-tuning (HCCF), as detailed in Table 3. In contrast to the traditional training using cross-entropy loss (flat softmax (Bertinetto et al., 2020) combined with negative log-likelihood), our HCCF incorporates several enhancements:

Table 3: Ablation Study of Hierarchical Cross-Modal Fine-Tuning (HCCF) on iNat18. This study highlights three key modifications from the Cross-Entropy (CE) loss baseline to our HCCF: using CLIP pre-trained text encoder (text embedding), hierarchical training (L67 and L123456), and contrastive loss (CL). The adoption of the CLIP pre-trained text encoder markedly boosts model performance, with hierarchical training and contrastive loss providing additional enhancements. For a comprehensive explanation, refer to Sec. 4.3.

| Models | AP | AC | R@90C | R@95C | Majority F1 | Leaf F1 | Leaf Top1 |
|---|---|---|---|---|---|---|---|
| CE loss baseline (Bertinetto et al., 2020) | 61.18 | 58.94 | 45.44 | 37.58 | 64.27 | 64.57 | 47.33 |
| CE loss + text embedding | 66.25 | 64.09 | 51.72 | 43.66 | 69.42 | 69.31 | 53.10 |
| CE loss + text embedding + L67 | 67.81 | 65.7 | 54.13 | 46.09 | 70.81 | 70.66 | 54.07 |
| CE loss + text embedding + L1234567 | 69.18 | 67.07 | 56.32 | 48.28 | 71.99 | 71.81 | 53.68 |
| CL + text encoder + L1234567 (HCCF) | 72.75 | 70.60 | 59.56 | 52.60 | 72.73 | 75.16 | 55.78 |

Table 4: An ablation study of Hierarchical Cross-Modal Fine-Tuning (HCCF) over different training levels on iNat18 reveals intriguing insights. While training across more levels consistently enhances all metrics under CE loss, as illustrated in Table 3, the same doesn't hold true for contrastive loss. Training at the leaf level (denoted as L7) yields the highest leaf Top1 accuracy but falls short in hierarchical metrics compared to multi-level encoder head training. For metrics like AP, AC, and Leaf F1, comprehensive training across all levels (denoted as L123467) outperforms other configurations. Training on levels 6 and 7 alone achieves the peak for R@90C and R@95C. Broadening the training levels benefits hierarchical metrics, with the coarsest (level 1) and sub-finest (level 6) levels proving most advantageous. It's noteworthy that these findings diverge from the prevailing belief that top-1 accuracy benchmarks align with hierarchical metric rankings (Russakovsky et al., 2015), underscoring the importance of studying hierarchical metrics.

| Model | AP | AC | R@90C | R@95C | Majority F1 | Leaf F1 | Leaf Top1 |
|---|---|---|---|---|---|---|---|
| HCCF L7 | 72.40 | 70.33 | 59.36 | 52.42 | 72.33 | 74.72 | **56.69** |
| HCCF L67 | 72.64 | 70.65 | **60.53** | **53.22** | **72.85** | 74.88 | 56.10 |
| HCCF L567 | 72.62 | 70.51 | 59.69 | 52.92 | 72.72 | 74.97 | 55.80 |
| HCCF L4567 | 72.50 | 70.34 | 59.26 | 52.29 | 72.58 | 74.89 | 55.43 |
| HCCF L34567 | 72.52 | 70.36 | 59.46 | 52.27 | 72.65 | 74.87 | 55.29 |
| HCCF L234567 | 72.55 | 70.37 | 59.38 | 51.63 | 72.45 | 74.98 | 55.72 |
| HCCF L1234567 | **72.75** | **70.60** | 59.56 | 52.60 | 72.73 | **75.16** | 55.78 |

**Use of CLIP pre-trained text encoder**: To assess the benefits of the CLIP pre-trained text encoder, we modified the initial weights of the final fully connected layer in CE loss training by incorporating the CLIP pre-trained text embeddings for each category. This strategy harnesses the knowledge from the cross-modal pre-training set, creating a more optimized initial embedding space for the categories. This straightforward adjustment leads to a marked improvement in the CE baseline performance. While the effectiveness of leveraging the CLIP pre-trained encoder has been previously noted in contexts like few-shot classification (Xiao et al., 2022) and object detection (Jin et al., 2021), our work stands out as the first to apply this technique to hierarchical classification, achieving notable gains.

**Hierarchical training**: Unlike the flat softmax which aggregates the probabilities of child nodes to determine the mid-level node probability, our hierarchical training instructs the model to directly estimate the probability for each mid-level node. This strategy aims to better delineate the mid-level manifolds, as depicted in Fig. 1. This method further enhances performance, particularly in hierarchical metrics.

**Incorporation of contrastive loss**: As discussed in Sec. 3.2, the addition of the contrastive loss further augments the model's performance.

In summary, our HCCF approach, with its multiple enhancements, demonstrates superior performance compared to traditional training methods. We additionally performed hierarchical cross-modal fine-tuning at various levels, beginning exclusively with the leaf level and culminating with all levels. As indicated in Table 4, the utilization of all levels yielded the optimal hierarchical performance. However, it adversely affected the leaf-level performance. Harnessing the bottom two levels proved to be the most cost-efficient strategy. Intriguingly, incorporating additional levels, such as levels 5, 6, and 7, did not improve performance compared to just using levels 6 and 7. It's noteworthy that these findings diverge from the prevailing belief that top-1 accuracy benchmarks align with hierarchical metric rankings (Russakovsky et al., 2015), underscoring the importance of studying hierarchical metrics.

Table 5: Evaluation of our cutting-edge diffusion-based inference against established state-of-the-art (SOTA) methods on iNat18. Despite all inference techniques utilizing the same trained model, our diffusion and differentiable diffusion approaches surpass all the SOTA methods. Notably, this enhancement is achieved without any modifications to the trained model.

| Model | AP | AC | R@90C | R@95C | Majority F1 | Leaf F1 | Leaf Top1 |
|---|---|---|---|---|---|---|---|
| Top-down (Redmon & Farhadi, 2017) | 64.36 | 61.72 | 46.10 | 34.97 | 68.54 | 68.36 | 46.62 |
| Advanced-top-down (Jain et al., 2023) | 72.11 | 69.98 | 58.09 | 46.96 | 76.23 | 75.96 | 55.71 |
| Bottom-up (Valmadre, 2022) | 72.75 | 70.60 | 59.56 | 52.60 | 72.73 | 75.16 | 55.78 |
| Diffusion (Ours) | 73.48 | 71.88 | **62.48** | **55.53** | 75.94 | 75.71 | 56.33 |
| Differentiable diffusion (Ours) | **73.82** | **71.91** | 61.99 | 53.36 | **76.01** | **76.09** | **59.70** |

Table 6: Our diffusion-based inference method is model-agnostic, enhancing classifier performance across all metrics. This improvement is consistent whether the model is trained comprehensively across all levels (HCCF L123456), on level 6 and level 7 (HCCF L67), or solely at the leaf level (Flat softmax).

| Model | AP | AC | R@90C | R@95C | Maj F1 | Leaf F1 | Leaf Top1 |
|---|---|---|---|---|---|---|---|
| HCCF L1234567 bottom-up | 72.75 | 70.60 | 59.56 | 52.60 | 72.73 | 75.16 | 55.78 |
| HCCF L1234567 diffusion | 73.60 | 71.85 | 62.06 | 54.97 | 74.79 | 75.82 | 56.50 |
| HCCF L1234567 differentiable diffusion | 73.82 | 71.91 | 61.99 | 53.36 | 76.01 | 76.09 | 59.70 |
| HCCF L67 bottom-up | 72.64 | 70.65 | 60.53 | 53.22 | 72.85 | 74.88 | 56.10 |
| HCCF L67 diffusion | 73.35 | 71.63 | 62.26 | 55.25 | 74.57 | 75.51 | 56.84 |
| HCCF L67 differentiable diffusion | 73.23 | 71.37 | 61.38 | 53.30 | 75.48 | 75.44 | 59.51 |
| Flat softmax bottom-up | 69.18 | 67.07 | 56.32 | 48.28 | 71.99 | 71.81 | 53.68 |
| Flat softmax diffusion | 69.45 | 67.56 | 56.47 | 48.61 | 72.57 | 72.31 | 54.14 |
| Flat softmax differentiable diffusion | 69.20 | 67.12 | 56.40 | 48.75 | 71.96 | 71.81 | 53.84 |

### 4.4 COMPARE DIFFUSION WITH OTHER INFERENCE METHODS

In addition to training, inference plays a pivotal role in hierarchical classification for final decision-making. We evaluated our innovative diffusion-based techniques, including both general and differentiable diffusion, against traditional top-down and bottom-up inference methods. The results, presented in Table 5, reveal that our methods notably surpass existing ones. Intriguingly, diffusion not only enhances hierarchical metrics but also boosts the leaf-level top 1 accuracy. The fact that our general diffusion doesn't necessitate extra training makes this discovery particularly noteworthy. When trained using our differentiable diffusion, the performance escalates even further.

Differentiable diffusion excels in numerous metrics over general diffusion except in R@90C and R@95C. The advantage of general diffusion is its simplicity and the absence of a training requirement. Further experiments, as seen in Table 6, confirm the consistency of these findings across various models. This underscores the novelty and success of our diffusion-centric approach to classification.

### 4.5 SOCIAL IMPACT AND LIMITATIONS

Our research introduces innovative training methodologies and novel diffusion mechanisms for hierarchical classification. Extensive experiments show that our proposed methods deliver more accurate and impactful hierarchical classification results. These advancements have potential implications for various applications, from object detection to the realm of autonomous driving. While our techniques represent a significant leap forward, they have limitations. Our empirical evaluations have been primarily anchored to the well-structured iNat18 and iNat21 datasets. As a next step, it would be pivotal to assess the versatility of our method in diverse real-world contexts, including its potential role in autonomous driving systems.

## 5 CONCLUSIONS

This paper introduces a fresh perspective on the hierarchical classification problem by viewing it through the lens of manifold learning. Leveraging this approach, we present innovative strategies for training and inference. Our proposed hierarchical cross-modal contrastive loss and graph-based diffusion methods for hierarchical predictions offer a nuanced balance between coarse and fine-class predictions. Evaluations on iNat18 and iNat21 datasets demonstrate the superior performance of our methods in terms of both top-1 accuracy and various hierarchical metrics, marking a notable advancement in the field of hierarchical classification.

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
