# OpenReview forum: "Hierarchical Classification by Training to Diffuse on the Manifold"
_ICLR.cc/2024/Conference — ICLR 2024 Conference Withdrawn Submission_

### Official Review · Reviewer_uzTo · 2023-10-30

**Soundness:** 2 fair
**Presentation:** 3 good
**Contribution:** 2 fair
**Rating:** 3
**Confidence:** 4

**Summary:**

This paper aims to address hierarchical classification by regularzing the CLIP-based pre-training methods using the given hierarchical taxonomy. Specifically, the authors propose a hierarchical cross-modal contrastive loss that computes contrative losses w.r.t labels at hierarchical levels in the taxonomy during the training stage, and a diffusion-based methodology to further adjust the posteriors output by the classification model during the test stage. To demostrate the effectiveness of the proposed methods, the authors present various results on iNat18 and iNat21 datasets.

**Strengths:**

+ Hierarchical classification is an interesting topic in machine learning, and hierarchical classification based on regularizing the CLIP-based pre-training methods has not been seen in the previous literature.
+ The paper is well-organized  and well-written except for the proposed differentiable diffusion strategy in Line 221-228.
+ The authors conduct experiments on the popular datasets iNat18 and iNat21.

**Weaknesses:**

+ The proposed hierarchical cross-modal contrastive fine-tuning loss is a straightforward extension based on CLIP, and there is a lack of explanation and theoretical justification for why this model is effective for hierarchical classification from a manifold learning perspective.

+ The description of the proposed differentiable diffusion strategy in Lines 221-228 is unclear. For example, how can we train a linear mapping to replace the one optimized by Eq. (6)?

+ In Lines 269-270, the authors claim that '... we strictly adhered to the code provided by Valmadre [1]. Our results for iNat21 are consistent with those presented by Valmadre [1].' However, this paper uses a more powerful backbone, ResNet-50, in contrast to the ResNet-18 adopted in [1]. The puzzling aspect is that this paper reports much lower performance on iNat21 compared to Valmadre [1]. What is the reason for such a performance degradation?

+ The main contribution of this paper is a diffusion-based inference. In my view, it's a general method, so it can also be used in ImageNet-based fine-tuning methods. Why not use it directly in ImageNet-based methods, similar to Valmadre [1]?

+ Eq. (4) introduces a crucial hyper-parameter as it plays an important role in injecting prior information from the connection matrix into the final prediction. However, there is no discussion about it in the experimental section or from a theoretical perspective.

+ There are several typos, such as 'the neural network Remarkably' in Line 185, and Equation (6) in Line 218 and Eq. (6) in Line 221.

**Questions:**

Please refer to the weakness section.

---

### Official Review · Reviewer_ijE5 · 2023-11-01

**Soundness:** 2 fair
**Presentation:** 3 good
**Contribution:** 3 good
**Rating:** 5
**Confidence:** 3

**Summary:**

This paper proposes methods to improve hierarchical classification in the fine-tuning and inference stages.  A hierarchical cross-modal contrastive finetuning strategy (HCCF) is proposed for fine-tuning the model. Diffusion-based inference methods are proposed to exploit posteriors at multiple levels. Experiments on iNat-18 and iNat-21 datasets show the proposed methods outperform state-of-the-art methods on hierarchical classification.

**Strengths:**

- The proposed HCCF achieves better results than all compared soft-max-based hierarchical fine-tuning methods in hierarchical classification.
- Ablation studies are conducted on the effects of CLIP pre-trained text conder, hierarchical training, and contrastive training.
- The diffusion-based inference method does not require training and improves the performance of hierarchical classification.
- The proposed diffusion-based inference method outperforms existing inference methods for hierarchies.

**Weaknesses:**

- HCCF loss trained with only L7 in Table 4 surpasses the existing loss in Table 1 largely. These results show the significant gain of HCCF is obtained mainly by contrastive training rather than considering the hierarchy. Contrastive finetuning of CLIP seems to be inspired by (Goyal et al., 2023).

- Some details are unclear. See the Questions section below.

Minor problems.
- Line 132. The two terms $log()$ are the same. This seems to be a typo.
- In Sec.3.2 $f$ is image feature embedding. In Sec.3.2 $f$, is a prediction output (softmax?). Using the same notation would be confusing.
- In Table 5, Advanced-top-down performs the best in Majority F1; however, the number is not bold.

**Questions:**

- In my understanding, the level-specific head embeds an image into different feature spaces in different category levels. What are the features in Fig.1 represent? Actual feature visualization, such as t-SNE, may also improve persuasiveness.

- The parameter $\alpha$ needs to be set for the diffusion-based inference. How this value is set needs to be clarified. The impact of this value also should be evaluated.

- The details of differential diffusion is not show. This should be shown.

---

### Official Review · Reviewer_fjJY · 2023-11-07

**Soundness:** 3 good
**Presentation:** 3 good
**Contribution:** 3 good
**Rating:** 6
**Confidence:** 4

**Summary:**

This paper proposes two techniques for improving hierarchical image classification: a training method and an inference method. The training approach provides better utilization of pre-trained models by adopting an image-text model (CLIP) and applying a contrastive loss based on the (fixed) label embeddings from the text encoder. The hierarchical contrastive loss comprises an independent sum over per-level contrastive losses. The inference procedure applies diffusion to iteratively propagate scores in the (undirected) graph defined by the taxonomy. Using the iNaturalist datasets (2018, 2021-mini), the training method is shown to obtain greatly improved results over simply fine-tuning the CLIP image encoder using existing losses. The inference algorithm is also demonstrated to further improve the results. The authors also propose a differentiable diffusion approach, which replaces the adjacency matrix with a learned matrix.

**Strengths:**

1. The baseline methods were evaluated using the same pre-trained initialization (CLIP image encoder) and a large improvement is achieved (+8 points Average Precision).
1. The ablative study in Table 3 shows the improvement in accuracy obtained by using the text embedding and the contrastive loss.
1. The inference procedure was compared to other approaches using the same model in Table 5, and with different models in Table 6.
1. Good use of hierarchical metrics.
1. The methods are demonstrated on both balanced and imbalanced datasets.
1. Overall, the ablative studies were well-constructed, and answered most of my queries about the method.

**Weaknesses:**

I will use [1] to denote Valmadre (2022).

1. What is "hierarchical training" with CE loss in the ablative study? (L294; also "CE loss + Lxxxx" distinct from "Contrastive Loss + Lxxxx" in Table 3) This needs to be explicitly defined.
1. The proposed level-wise loss does not seem to produce consistent probabilities as defined in [1]; i.e. where the probability of an internal node is greater than or equal to the sum of its children.
1. The level-wise loss might not apply as well in hierarchies that contain leaf nodes of different depths (I believe the iNaturalist taxonomy has a uniform depth of 7).
1. The results in Table 4 lack sufficient significance to conclude that "for metrics like AP, AC, and Leaf F1, comprehensive training across all levels (denoted as L123467) outperforms other configurations" and consequently that "these findings diverge from the
prevailing belief that top-1 accuracy benchmarks align with hierarchical metric rankings". Most of these numbers seem about the same, with the possible exception of Leaf F1. (However, I agree that L67 and L7 are significantly better for R@xxC and Leaf Top1.)
1. While the ablative study shows some important results, I would have liked to see further investigation into the importance of the image encoder. In particular, while the authors used CLIP pretraining for the baselines, the numbers are quite close to those of [1], which used standard ImageNet pretraining. I think it would be useful to highlight this in the paper (it shows that the method better leverages the pretrained model). Furthermore, since the label embeddings seem crucially important, it would be interesting to investigate whether the CLIP image encoder is even necessary by using ImageNet pretraining with the same label embeddings.
1. It would be useful to generate the operating curves as in [1], to highlight whether the method achieves a better trade-off or simply dominates the other methods.
1. No confidence intervals (error-bars).
1. The diffusion procedure was not evaluated on models that were trained using the "parameter sharing softmax" and "soft-max-margin" losses in Table 6.
1. It's inaccurate to characterize the *inference* procedure of [1] as "bottom-up" (e.g. Table 5): it returns the most-informative ($\approx$ deepest) label that exceeds a confidence threshold. In fact, if the threshold is > 0.5 and the probabilities satisfy the tree constraint, then it is equivalent to greedy top-down (since there can only be one path with confidence > 0.5). A better example of bottom-up inference would be Davis et al. "Hierarchical Semantic Labeling with Adaptive Confidence", where the inference algorithm starts at the argmax leaf node and proceeds upwards.
1. Several details are unclear, see questions.

Minor:
1. It seems as though the "Softmargin" and "Descendant softmax" rows might be interchanged between Table 1 and Table 2? The results of [1] showed the "softmargin" to perform consistently better than "descendant softmax", as in Table 2 but not in Table 1.
1. In (1), it seems that one of the summations should be over all leaf nodes, rather than $L(y)$.
1. The "parameter-sharing softmax" of Salakhutdinov et al. (2011), also considered in [1], is effectively a hybrid method: logits are obtained in a top-down fashion, normalization is performed at the leaf nodes, and then probabilities are obtained bottom-up.

Suggestions:
1. I would use terminology like "graph diffusion" at the start of the paper to make it clear that you're not considering generative diffusion (DDPMs, stable diffusion, etc.).
1. "down-top" should be "bottom-up", unless you mean something specific by this?

**Questions:**

1. Are the summations over $i$ and $j$ in (2) over all labels in level $l$ (in which case, it should be $N_l$ rather than $N$?), or all labels at all levels (in which case, are the ancestor labels excluded?)?
1. How would the loss in (2) be extended to taxonomies where not all leaf nodes occur at the same depth?
1. Is the diffusion process applied to raw image-text similarities, or are the probabilities normalized; if so, are they normalized globally or per-level (as appears to be the case in the loss)?
1. Which model was used to produce Table 5?
1. What explains the difference between the results for "Flat softmax" in Table 2 (61.18 AP) and Table 6 (69.18 AP)?